# Effect of Voltage on the Microstructure and High-Temperature Oxidation Resistance of Micro-Arc Oxidation Coatings on AlTiCrVZr Refractory High-Entropy Alloy

**Zhao Wang** [1], **Zhaohui Cheng** [2], **Yong Zhang** [2], **Xiaoqian Shi** [2,*], **Mosong Rao** [2] and **Shangkun Wu** [2]

[1] Shaanxi Provincial Office of Defense Science, Technology and Industry, Xi'an 710021, China
[2] School of Materials Science and Chemical Engineering, Xi'an Technological University, Xi'an 710021, China
* Correspondence: shixiaoqian0220@163.com

**Abstract:** In order to improve the high-temperature oxidation resistance of refractory high-entropy alloys (RHEAs), we used micro-arc oxidation (MAO) technology to prepare ceramic coatings on AlTiCrVZr alloy, and the effects of voltage on the microstructure and high-temperature oxidation resistance of the coatings were studied. In this paper, the MAO voltage was adjusted to 360 V, 390 V, 420 V, and 450 V. The microstructure, elements distribution, chemical composition, and surface roughness of the coatings were studied by scanning electron microscopy (SEM), energy dispersive (EDS), X-ray photoelectron spectroscopy (XPS), and white-light interferometry. The matrix alloy and MAO-coated samples were oxidized at 800 °C for 5 h and 20 h to study their high-temperature oxidation resistance. The results showed that as the voltage increased, the MAO coating gradually became smooth and dense, the surface roughness decreased, and the coating thickness increased. The substrate elements and solute ions in the electrolyte participated in the coating formation reaction, and the coating composition was dominated by $Al_2O_3$, $TiO_2$, $Cr_2O_3$, $V_2O_5$, $ZrO_2$, and $SiO_2$. Compared with the substrate alloy, the high-temperature oxidation resistance of the MAO-coated samples prepared at different voltages was improved after oxidation at 800 °C, and the coating prepared at 420 V showed the best high-temperature oxidation resistance after oxidation for 20 h. In short, MAO coatings can prevent the diffusion of O elements into the substrate and the volatilization of $V_2O_5$, which improves the high-temperature oxidation resistance of AlTiCrVZr RHEAs.

**Keywords:** refractory high-entropy alloys; micro-arc oxidation; voltage; microstructure; high-temperature oxidation resistance



## 1. Introduction

With the development of aerospace technology, the requirement for the high-temperature performance of components is increasing. At present, most of the materials used in the high-temperature field are nickel-based superalloys, but the application temperature is only between 116 °C and 1277 °C, which have been unable to meet the demand [1,2]. Refractory high-entropy alloys (RHEAs) composed of three or more refractory elements (Ti, V, Cr, Zr, Nb, Mo, Hf, Ta, and W), and other elements in approximate molar ratios, have the potential to be applied to high-temperature structural parts at higher temperatures because of their excellent room-temperature and high-temperature mechanical properties [3–5]. For example, the yield strength of NbMoTaW and VNbMoTaW RHEAs at 1600 °C exceeds 400 MPa, far exceeding the yield strength of Inconel 718 nickel-based superalloy at 1000 °C of 200 MPa. However, refractory high-entropy alloys tend to rapidly form volatile oxides ($MoO_3$, $WO_3$, $V_2O_5$, etc.) and easily spalling oxides ($Nb_2O_5$ and $Ta_2O_5$) that do not have high-temperature protection capabilities at high temperatures, resulting in high levels of oxidation in high-temperature aerobic environments [6–9]. Poor high-temperature oxidation resistance has become one of the key factors restricting the application of refractory

high-entropy alloys in high-temperature fields, and it has become particularly urgent to improve the high-temperature oxidation resistance of RHEAs.

Micro-arc oxidation (MAO) is a surface modification technology with simple operation, environmental protection, and high efficiency. The ceramic coating formed has the characteristics of strong adhesion to the matrix and uniform growth of the coating, along with the ability to improve the high-temperature oxidation resistance of the matrix metal [10–12]. It was found that after the preparation of the MAO coating dominated by $Al_2O_3$ on the surface of the Ti-45Al-8.5Nb alloy and oxidation at 900 °C for 100 h, the minimum weight gain was only 0.396 $g/m^2$, which significantly improved the high-temperature oxidation resistance [13]. Of course, some progress has also been made in the study of MAO technology to improve the high-temperature oxidation resistance of $AlTiNbMo_{0.5}Ta_{0.5}Zr$ and AlTiCrVZr refractory high-entropy alloys [10,14]. However, the current research on the use of MAO technology to improve the high-temperature oxidation resistance of RHEAs is still in its initial stage. Voltage as one of the key influencing factors of MAO technology plays a decisive role in the electric field strength in the MAO process, affects the migration rate of anions and ions, and then affects the growth rate, density, and bonding strength of the coating [15–18]. The thickness, density, and bonding strength of the coating are closely related to the high-temperature oxidation resistance of the coating. At present, the effect of MAO voltage on the high-temperature oxidation resistance of RHEAs has not been reported.

In this paper, by adjusting the voltage, the MAO coating was prepared in situ on the AlTiCrVZr RHEA. The effects of voltage on the micromorphology, roughness, chemical composition, and high-temperature oxidation resistance of the surface and cross-section of the MAO coating were studied. These research results can promote the development of MAO technology and RHEAs, and provide a reference for the surface modification of RHEAs applied in high-temperature environments.

## 2. Experimental Process

### 2.1. Sample Pretreatment

The purchased commercial AlTiCrVZr RHEA was processed into blocks with a size of 7 mm × 7 mm × 5 mm, and all samples were ground step by step with 240#~2000# sand paper, and then sonicated in anhydrous ethanol for 20 min, dried in air, and set aside.

### 2.2. Preparation of the MAO Coating

Using the MAO-20H power supply independently developed by Xi'an Technological University, the working voltages were 360 V, 390 V, 420 V, and 450 V, and the MAO coating was prepared on the surface of the AlTiCrVZr high-entropy alloy. The electrolyte was 50 g/L $Na_2SiO_3$ + 25 g/L $(NaPO_3)_6$ + 5 g/L NaOH, the frequency was 600 Hz, the duty cycle was 8%, the MAO treatment lasted 5 min, and the electrolyte temperature remained between 30 °C and 35 °C. The samples treated by MAO were sonicated in anhydrous ethanol and dried for further study.

### 2.3. Microstructure Analysis and Performance Characterization

Scanning electron microscopy (SEM, TESCAN VEGA 3-SBH, Taseken Trading Company, Shanghai, China) was used to observe the surface and cross-sectional morphology of the coating, and the elemental content of the coating was studied with an equipped energy-dispersive X-ray spectrometer (EDS, TESCAN VEGA 3-SBH, Taseken Trading Company, Shanghai, China). The white-light interferometer (ZeGage, ZYGO, Connecticut, Middlefield, CT, USA) was used to obtain the surface 3D morphology of the coating, and the surface roughness analysis was performed. An Axis Ultra DLD X-ray Photoelectron Spectrometer (XPS) (Shimadzu Corporation, Hadano, Japan) was used to analyze the bond composition. The experimental temperature of the high-temperature oxidation resistance test was 800 °C, and the experimental time was 5 h and 20 h. The matrix and the samples prepared using different voltages were placed in an alumina crucible and then placed in

a high-temperature chamber furnace where the temperature had risen to 800 °C for the oxidation test. After oxidation at 800 °C, the cross-sectional morphology of the coating was observed by SEM, and the diffusion of the elements was analyzed by EDS.

## 3. Results and Discussion

### 3.1. Microstructure of the Coating

Figure 1 shows the surface morphologies of the MAO coatings prepared using different voltages, and there are obvious pores on the coating, showing a typical "volcano crater" morphology. During the growth of the coating, the presence of holes as discharge channels is unavoidable, and the formation of holes is mainly attributed to the outward escape of internal molten oxides and gases [10]. As the voltage increased, less particulate matter accumulated on the coating. This is because the higher the voltage, the more intense the reaction; the generated molten particles will be connected to each other, and it is difficult to observe the obvious particle aggregation phenomenon. In addition, the higher the voltage, the more energy and heat are released during the MAO process, and the more melt is generated, which will cover the uneven area formed in the early stage, making the coating smoother and smoother. Of course, higher voltages can also cause greater thermal stress in the melt during the electrolyte cooling, resulting in more microcracks being observed on the coatings [19].

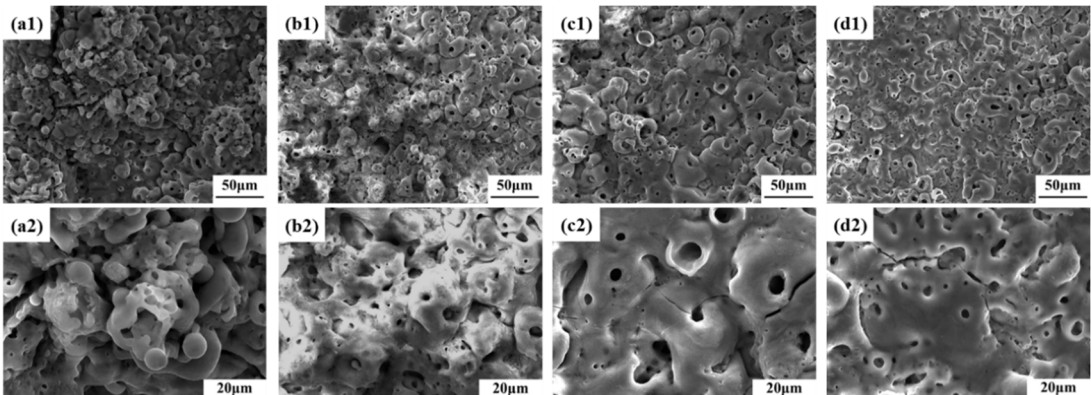

**Figure 1.** Surface morphologies of the coatings prepared using different voltages: (**a1**,**a2**) 360 V, (**b1**,**b2**) 390 V, (**c1**,**c2**) 420 V, and (**d1**,**d2**) 450 V.

The surface element content of the MAO coatings prepared using different voltages is shown in Table 1. The coating was dominated by O and Si elements and contained a small amount of P elements, all of which were derived from the electrolyte, and the content was related to their concentration. The matrix elements were all involved in the coating formation reaction, of which the content of Al, Ti, and Zr was relatively high, while the content of V and Cr was relatively small. The difference in elemental content is most likely related to the size of the oxygen affinity, and the greater the oxygen affinity, the more likely the element is to oxidize, and the higher the content. The less electronegative the element, the greater the affinity for oxygen [20,21]. The electronegative size of the five elements of Al, Ti, Cr, V, and Zr is ordered as: Zr < Ti < Al < V < Cr, so the content of Al, Ti, and Zr in the coating will be higher than the content of V and Cr [21]. With the increase in the MAO voltage, the content of O, Al, Ti, and Zr elements on the coating gradually decreased, while the content of Si and P elements increased. This may be because the probability of the matrix elements reaching the surface of the coating through the discharge channel decreases with the increase in the thickness of the coating; this is mainly applicable to the solute ions participating in the coating forming reaction.

**Table 1.** Surface element content of coatings prepared using different voltages (at.%).

| Voltages | O | Al | Si | P | Ti | Cr | V | Zr |
|---|---|---|---|---|---|---|---|---|
| 360 V | 71.9 | 3.0 | 11.9 | 1.1 | 3.1 | 2.3 | 1.3 | 5.4 |
| 390 V | 70.4 | 3.0 | 16.0 | 0.2 | 3.0 | 1.1 | 1.4 | 4.9 |
| 420 V | 70.6 | 2.9 | 16.3 | 0.4 | 2.7 | 1.0 | 1.4 | 4.7 |
| 450 V | 69.8 | 2.6 | 15.6 | 1.2 | 2.8 | 1.9 | 1.5 | 4.5 |

Figure 2 presents the 3D morphologies and surface roughness of the coatings prepared using different voltages, and the coating area selected for different test samples was 834.370 μm × 834.370 μm. There was no obvious trend observed regarding the influence of voltage on the surface roughness, but it is undeniable that the coating prepared at a low voltage was coarsest, while the surface roughness of the coating prepared at a high voltage was smallest. The surface roughness of the coating prepared at 360 V voltage was 10.299 μm, and the surface roughness of the coating prepared at 450 V voltage was 7.343 μm, which represents a large difference in roughness. The measurement results of the coatings surface roughness were consistent with those observed in Figure 1. The temperature of the micro-arc discharge area rose rapidly at higher working voltages, resulting in the formation of more molten oxides, which plays a role in repairing and leveling the coating.

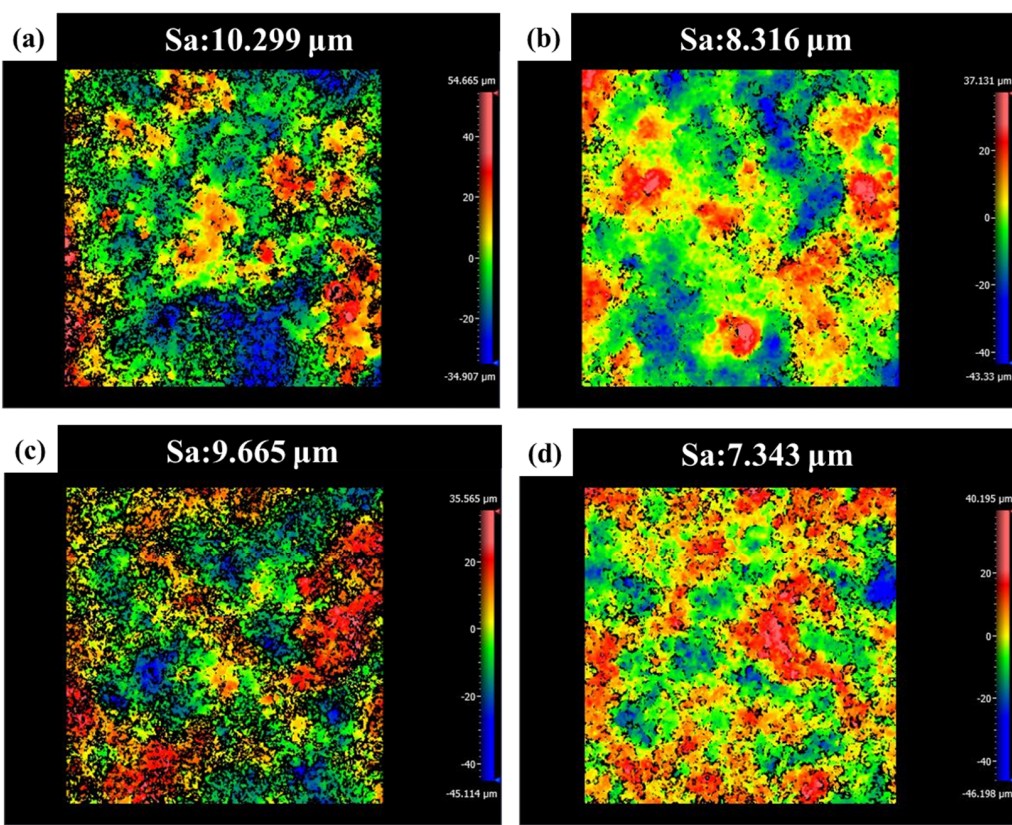

**Figure 2.** Two−dimensional morphologies of coatings prepared using different voltages: (**a**) 360 V, (**b**) 390 V, (**c**) 420 V, and (**d**) 450 V.

Cross-sectional morphologies of the coatings prepared using different voltages are shown in Figure 3. It can be seen that the coating and the matrix were well bonded, there was no obvious gap, and the interface junction was uneven. There were pores of different sizes at the cross-section of the coating, but these pores did not penetrate the entire coating. The thickness of the coating prepared at 360 V, 390 V, and 420 V did not differ much, all of which were around 40 μm, and the thickest coating prepared at 450 V was approximately

50 µm. In general, the higher the voltage, the thicker the MAO coating. Of course, this is not absolute, because the junction between the MAO coating and the matrix interface is uneven, which may lead to an inconsistent coating thickness between the two microregions. The higher the working voltage, the more intense the MAO reaction and the higher the growth rate of the coating, resulting in an increase in the thickness of the coating. Although the thicker coatings prepared at 450 V provide better protection for the matrix, larger cracks can be clearly observed in the cross-sectional topography that could cause the coating to peel off during high-temperature oxidation, which is extremely detrimental to improving high-temperature oxidation resistance.

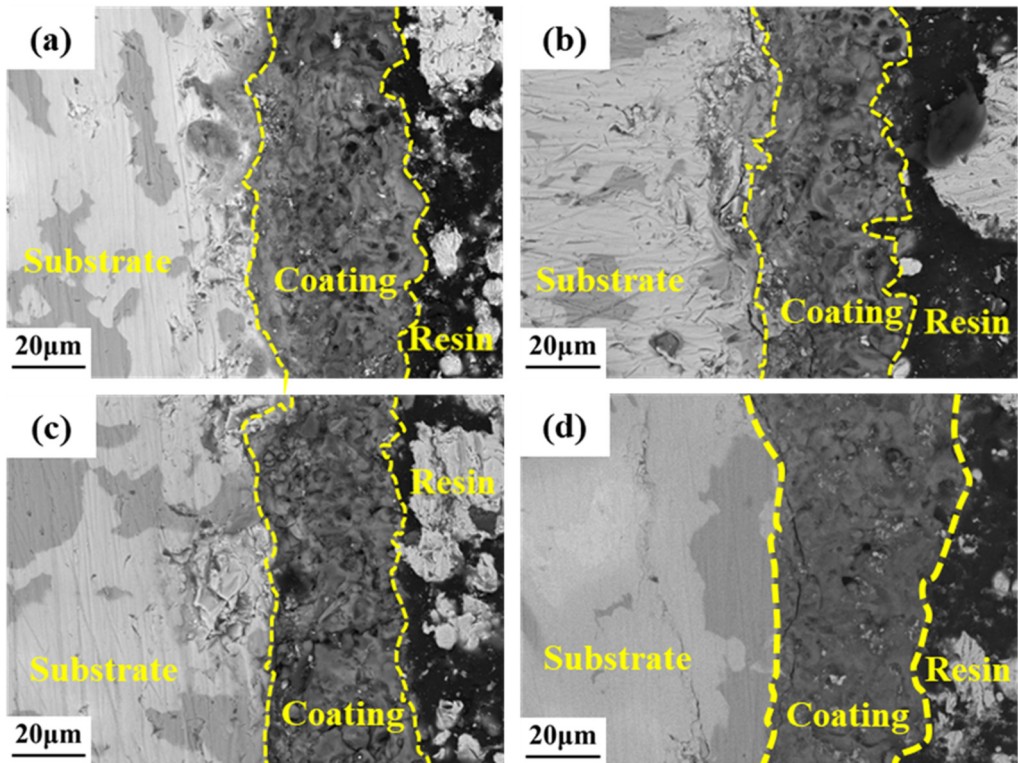

**Figure 3.** Cross-sectional morphologies of coatings prepared using different voltages: (**a**) 360 V, (**b**) 390 V, (**c**) 420 V, and (**d**) 450 V.

*3.2. Chemical Composition of the Coating*

In order to determine the presence of elements in the MAO coating on the AlTiCrVZr RHEA, XPS was used for analysis. The XPS full spectrum of the MAO coating prepared at 420 V is shown in Figure 4, and the eight peaks of Al2p, Si2p, Zr3d, C1s, Ti2p, V2p, O1s, and Cr2p were detected in the coating; of these, C1s was the source of contamination. It can be seen from the XPS full spectrum that the coating was mainly composed of O, Si, Al, Ti, Cr, V, and Zr elements, indicating that the solute elements and matrix elements in the electrolyte were involved in the coating-forming reaction.

The XPS high-resolution spectrum of each element on the coating prepared at 420 V is shown in Figure 5. There was only one peak in the Al2p spectrum, corresponding to a binding energy of 75.51 eV, indicating that the predominant form of Al present was $Al_2O_3$ [22]. The binding energies corresponding to the two peaks of Ti2p were 459.46 eV and 465.22 eV, indicating that Ti was present in the coating in the form of $TiO_2$ [23]. From the high-resolution spectrum of Cr2p, it can be seen that the binding of Cr2p had two peaks at 577.44 eV and 587.23 eV, corresponding to the Cr–O bond, representing $Cr_2O_3$ [10]. The peaks in the V2p spectrum were 517.99 eV and 523.87 eV, corresponding to the V–O bond, representing $V_2O_5$ [24]. The high-resolution spectrum of Zr3d showed that it had two peaks, corresponding to binding energies of 183.32 eV and 185.73 eV, and the surface

Zr present in the coating was $ZrO_2$ [25]. The peak of the Si2p spectrum was 103.10 eV, indicating that the Si present in the coating was $SiO_2$ [26].

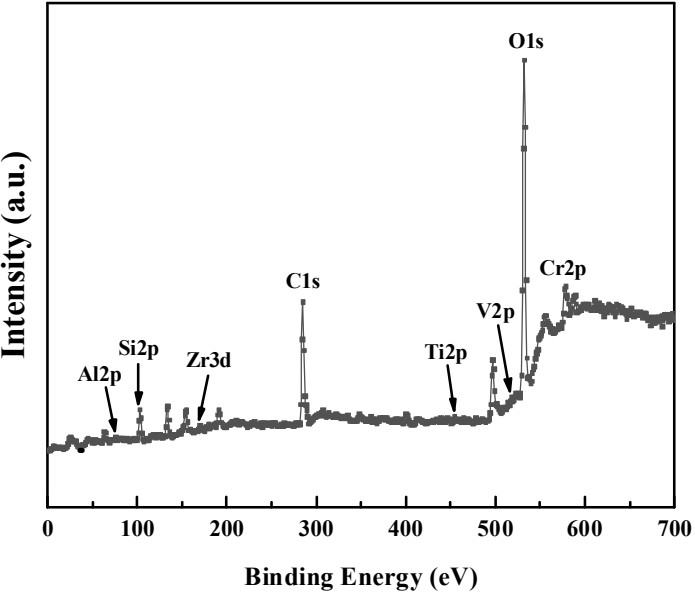

**Figure 4.** XPS full spectrum of the coating prepared at 420 V.

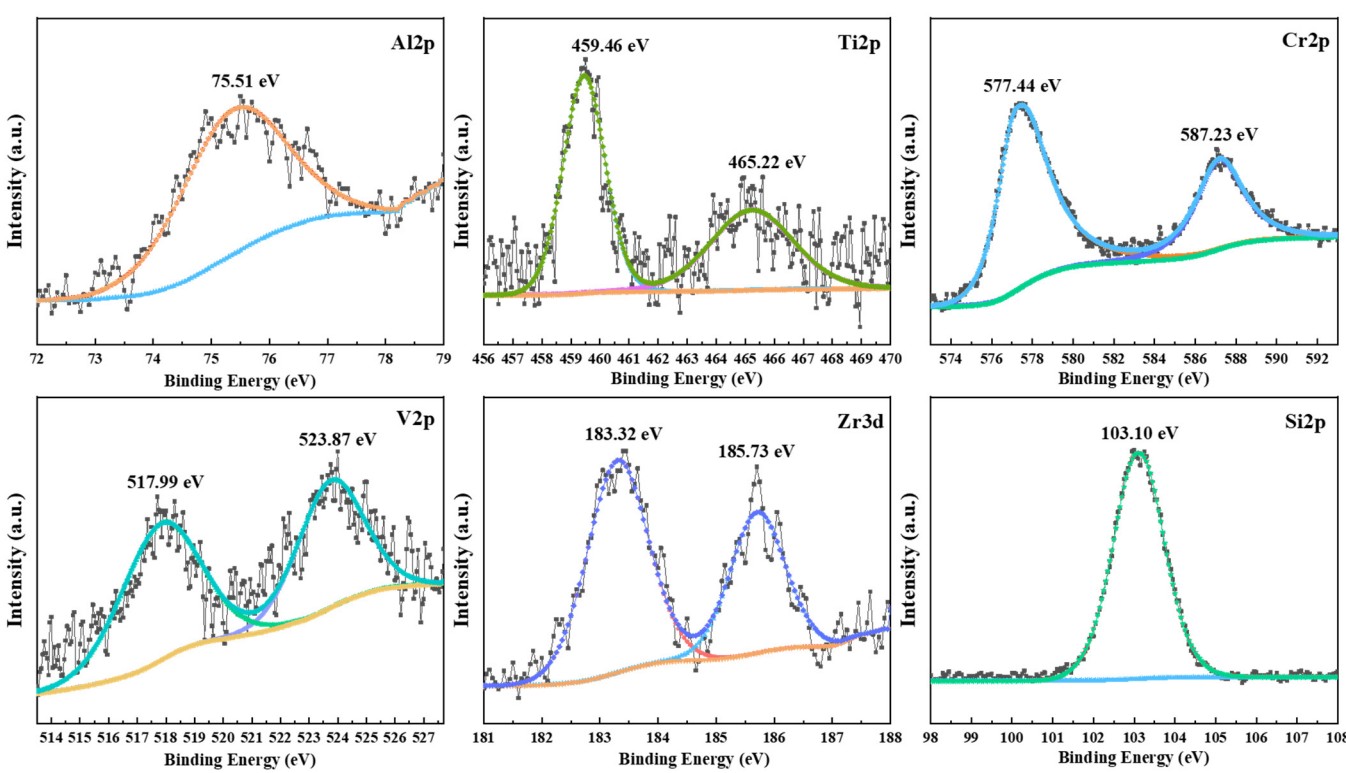

**Figure 5.** High-resolution spectrum of each element of the coating prepared at 420 V.

The matrix elements and solute ions in the electrolyte were involved in the coating-forming reaction, and the coating composition was dominated by $Al_2O_3$, $TiO_2$, $Cr_2O_3$, $V_2O_5$, $ZrO_2$, and $SiO_2$. Many studies have shown that $Al_2O_3$, $TiO_2$, $Cr_2O_3$, $ZrO_2$, and $SiO_2$ formed in the coating can prevent oxygen from diffusing directly into the matrix, thereby improving the high-temperature oxidation resistance of RHEAs [14,27]. We believe that the MAO coating prepared on AlTiCrVZr RHEA has the potential to improve the high-temperature oxidation resistance of matrix alloy.

*3.3. High-Temperature Oxidation Resistance*

Figure 6 shows the cross-sectional morphologies and elemental distribution of the MAO samples prepared using different voltages at 800 °C for 5 h. It can be seen that, after oxidation at 800 °C for 5 h, the thickness of the matrix diffusion layer exceeded 700 μm, which was significantly greater than the thickness of the diffusion layer of samples after MAO. In addition, we observed the presence of significant cracks and a small number of holes in the diffusion layer of the matrix. The generation of cracks and holes is attributed to thermal stress and the volatilization of oxides, respectively. The MAO samples prepared at 360 V, 420 V, and 450 V showed little difference in the thickness of the diffusion layer after high-temperature oxidation, of approximately 500 μm. The thickness of the diffusion layer of the MAO sample prepared at 390 V after high-temperature oxidation was only approximately 400 μm. Combined with the microstructure of the coating, it was found that the MAO sample prepared at 390 V exhibited good high-temperature oxidation resistance, which is closely related to the compactness of the coating and fewer microscopic defects.

The EDS results showed that the O element was enriched in the diffusion layer, and uncoated samples had a high content in the diffusion layer and a large amount inside the matrix, which indicates that the MAO coating hinders the diffusion of O elements to the inside of the matrix to a certain extent. In addition, the content of Ti, Al, Cr, and Zr elements in the diffusion layer after high-temperature oxidation of the MAO sample was lower than that of the matrix. The content of element V in the diffusion layer is always very low, because the $V_2O_5$ formed after oxidation is extremely volatile at 800 °C, resulting in a very low content of V elements and creating holes in the diffusion layer.

Figure 7 shows the cross-sectional morphologies and elemental distribution of the MAO samples prepared using different voltages and the matrix after oxidation at 800 °C for 20 h. After oxidation at 800 °C for 20 h, the thickness of the matrix diffusion layer exceeded 1500 μm, which was greater than the thickness of the diffusion layer of the MAO-coated samples. Although the MAO samples prepared at 390 V at 800 °C oxidation for 5 h showed good high-temperature oxidation resistance, the protection effect of the coating was poor when the high-temperature oxidation time increased to 20 h. The MAO sample prepared at 420 V had the smallest diffusion layer thickness, of approximately 800 μm, and the best high-temperature oxidation resistance. The high-temperature oxidation resistance of the coating prepared at 450 V is known to be poor as the micro-arc discharge reaction at higher voltages is violent, and the heat released results in additional cracks in the coating; $O_2$ very easily invades the inside of the matrix through cracks; therefore, the coating loses its protective effect on the matrix.

From the results of cross-sectional EDS after high-temperature oxidation, it can be seen that O elements were enriched in the diffusion layer, while the matrix elements were less abundant in the diffusion layer; in particular, the content of V elements was extremely low. With the increase in high-temperature oxidation time, $O_2$ gradually invaded the interior of the matrix and combined with V elements to form $V_2O_5$ at high temperatures. $V_2O_5$ is extremely volatile at 800 °C, resulting in an extremely low V content in the diffusion layer. Combined with the cross-sectional morphology and element distribution after 5 h of high-temperature oxidation, it can be seen that the MAO coating can hinder the diffusion of O elements to the interior of the matrix and the volatilization of $V_2O_5$ to a certain extent. The obstruction effect is not only related to the compactness of the coating, but also to the thickness and internal defects of the coating. The key to improving the high-temperature oxidation resistance of the MAO coating is to prepare a uniform and dense coating with good bonding with the matrix and to reduce cracks in the coating as much as possible. Increasing the thickness of the coating when the above conditions are met can more effectively hinder the contact of $O_2$ with the matrix metal, thereby significantly improving the high-temperature oxidation resistance of RHEAs.

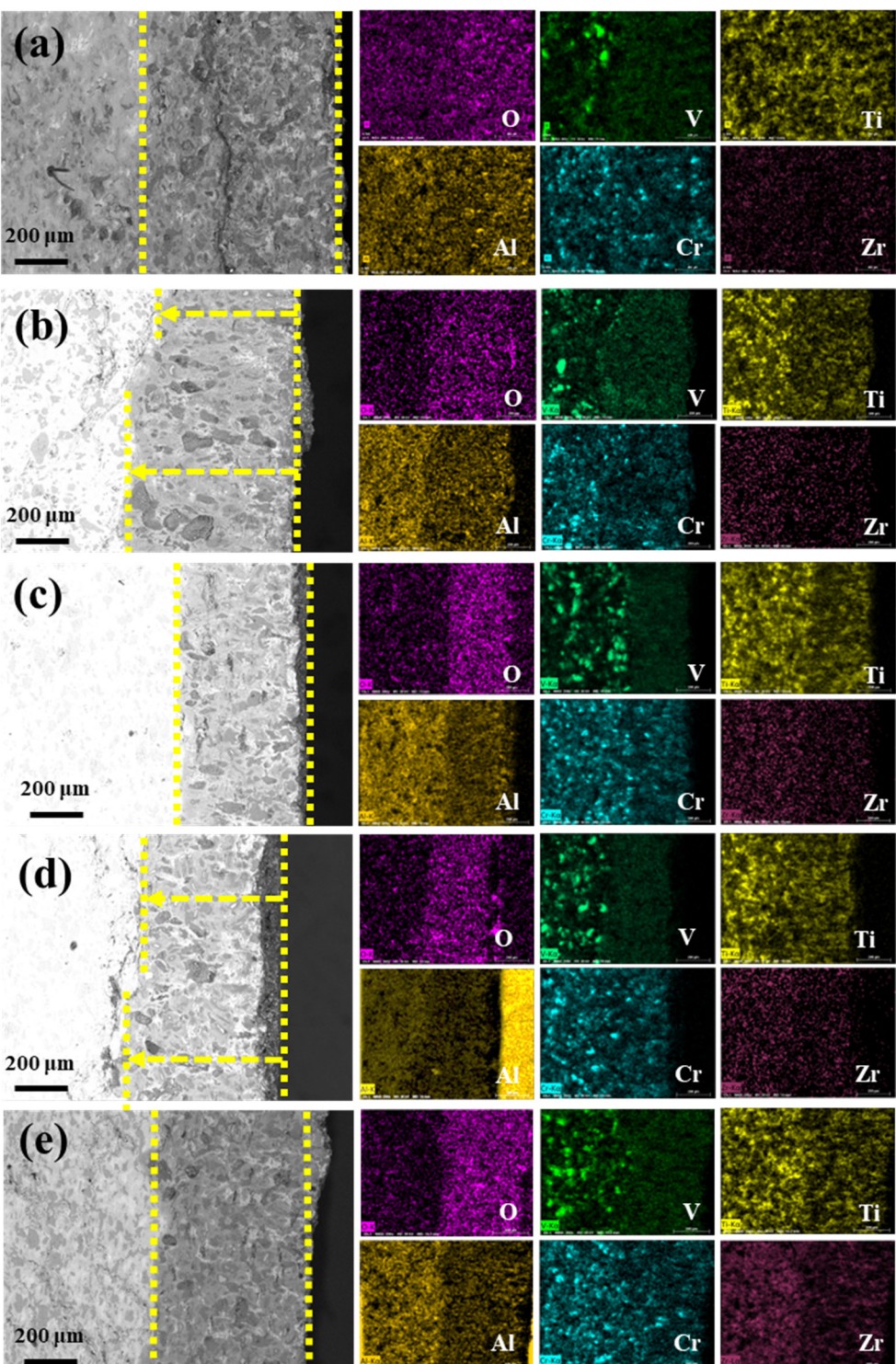

**Figure 6.** Cross-sectional morphologies and elements distribution of samples prepared using different voltages after oxidation at 800 °C for 5 h: (**a**) matrix, (**b**) 360 V, (**c**) 390 V, (**d**) 420 V, and (**e**) 450 V.



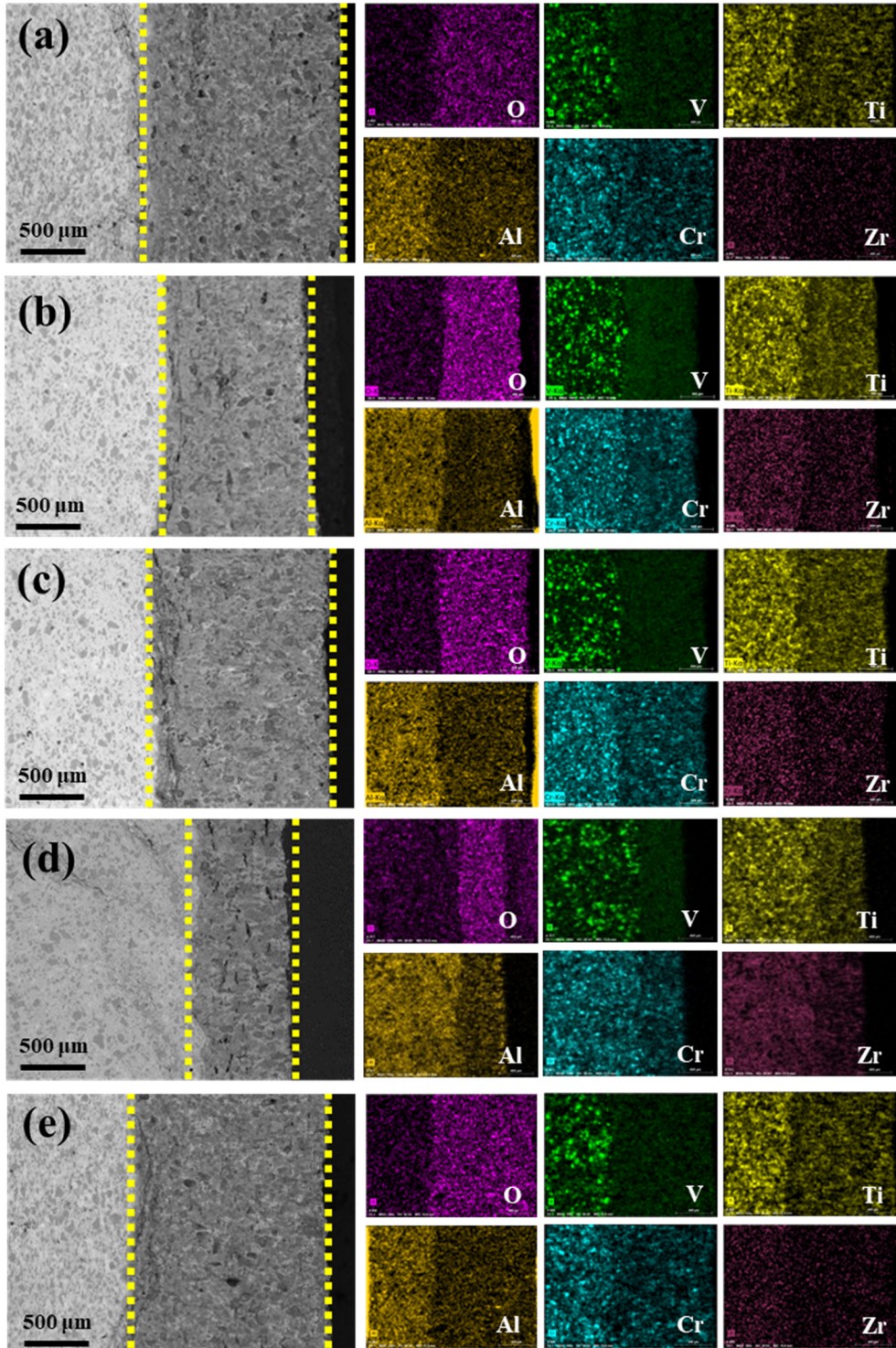

**Figure 7.** Cross-sectional morphologies and elements distribution of samples prepared using different voltages after oxidation at 800 °C for 20 h: (**a**) matrix, (**b**) 360 V, (**c**) 390 V, (**d**) 420 V, and (**e**) 450 V.

## 4. Conclusions

(1) Under different MAO voltages, ceramic coatings with a thickness of 40–50 μm were prepared on the AlTiCrVZr RHEA. With the increase in voltage, the surface of the coating was smoother and denser, but the internal defects of the ceramic coating increased, especially the obvious microcracks in the coating prepared at 450 V.

(2) The solute ions and matrix elements contained in the electrolyte during the MAO process were involved in the coating-forming reaction, and the coating composition was mainly $Al_2O_3$, $TiO_2$, $Cr_2O_3$, $V_2O_5$, $ZrO_2$, and $SiO_2$.

(3) Compared with the matrix alloy, the high-temperature oxidation resistance of MAO-coated samples prepared using different voltages was improved after 5 h and 20 h of oxidation at 800 °C. Among them, the coating prepared at 420 V exhibited better high-temperature oxidation resistance after long-term oxidation for 20 h.

**Author Contributions:** Conceptualization, Z.W. and X.S.; methodology, Z.C.; validation, Z.W. and Y.Z.; formal analysis, M.R.; investigation, S.W.; resources, Z.C.; data curation, Z.C.; writing—original draft preparation, Z.W.; writing—review and editing, X.S.; visualization, Y.Z.; supervision, M.R.; project administration, Y.Z.; funding acquisition, X.S. All authors have read and agreed to the published version of the manuscript.

**Funding:** This work was financially supported by the National Natural Science Foundation of China (No. 52071252) and the Key research and development plan of Shaanxi province industrial project (2021GY-208 and 2021ZDLSF03-11).

**Institutional Review Board Statement:** Not applicable.

**Informed Consent Statement:** Not applicable.

**Data Availability Statement:** Not applicable.

**Conflicts of Interest:** The authors declare that they have no known competing financial interests or personal relationships that could have appeared to influence the work reported in this paper.

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
