# Peer review of "Effect of Voltage on the Microstructure and High-Temperature Oxidation Resistance of Micro-Arc Oxidation Coatings on AlTiCrVZr Refractory High-Entropy Alloy"

_coatings, doi:10.3390/coatings13010014_

Round 1
Reviewer 1 Report
1. The authors evaluated the roughness of the coatings, which is an important parameter for coatings of this kind. However, it is not clear whether such roughness values are acceptable for coatings of this kind?
Thank you for your question. The roughness of the MAO coatings prepared in this paper was indeed large. However, for high temperature oxidation resistance, the roughness of the coating is not a decisive factor. The MAO coating is porous, and the large roughness is conducive to the release of thermal stress, thereby alleviating the peeling of the coating.
2. You can say that the applied voltage has no effect on the thickness of the films. Why? Need an explanation?
Thank you for your suggestions. In general, the voltage has an effect on the thickness of the coating. For example, Figure 3 showed that the coating prepared at 450V is thicker, mainly because the higher the working voltage, the more intense the MAO reaction and the higher the growth rate of the coating. Of course, this is not absolute, because the junction between the MAO coating and the matrix interface is uneven, which may lead to inconsistent coating thickness between the two microregions. We have explained this result in the article.
3. To what extent is the coating uniform in thickness over the entire area?
Thank you for your question. The thickness of the MAO coating is not uniform throughout the region, because the growth of the coating combines inward growth and outward growth, and the breakdown phenomenon always occurs preferentially in relatively weak areas. In general, the junction we observe at the interface of the coating and the matrix is uneven.
4. XPS is desirable for all voltages and also in coatings depth
Thank you for your question. Many studies have shown that voltage has little effect on the chemical composition of the MAO coating, so we only conducted XPS tests on the MAO coating prepared at 420V. In addition, the XPS test was not etched, we tested the surface chemistry of the coating.
Reviewer 2 Report
A very interesting paper about effects of different voltages in MAO coatings. The experiment presented in the paper is focused on AlTiCrVZr alloy. The experimental setup is clearly presented, methods properly explained and results properly analysed. Conclusions in the paper are really beneficial for all researchers in this field. Therefore, I recommended to the editor publishing your paper in its current form.
Thank you very much for your recognition, and we will continue to make efforts to research.
Reviewer 3 Report
1. The manuscript is generally well written and concerns issue important from the point of view of practical applications, namely metal alloys of good mechanical and anticorrosive properties at the high temperatures. Authors propose to use high entropy alloys (very popular object of investigations recently) for that purpose and they propose micro arc oxidation (again very popular object of investigations recently) as a method of surface treatment to increase the resistance to the high-temperature corrosion. They are right writing that “With the development of aerospace technology, the requirements for the high temperature performance of components are getting higher and higher, and traditional cobalt-based, nickel-based and other high temperature alloys have been difficult to meet the demand [1, 2].” Therefore, the reader may expect to find in this article the comparison of the properties of the newly-proposed alloys with traditional Ni and Co based superalloys. Nothing of this kind may be found in the considered manuscript.
Thanks for this suggestion of the reviewer, we have added the relevant content in the article and marked it with red font. In line 30 and 37 of the text, add “At present, most of the materials used in the high temperature field are nickel-based superalloys…”, “For example, the yield strength of NbMoTaW and VNbMoTaW RHEAs at 1600℃ exceeds 400 MPa…”.
2. From the very limited information on the corrosion resistance (authors investigated oxidation at one temperature and at two periods of time only) one cannot conclude if the corrosion resistance of the RHEA alloys is better than the corrosion resistance of the traditional Ni and Co based superalloys. Such comparison should be presented in the manuscript. If the anticorrosion properties of the new alloys are not better than the anticorrosion properties of the currently used materials there is no reason to investigate the new alloys. The content of the manuscript is rather limited. Authors investigated only one composition of the alloy, no information on the mechanical properties of the alloy may be found in the manuscript and the characterization of the corrosion resistance of the alloy is very limited.
Thank you for your question. I'm sorry that our research does have many shortcomings. MAO technology is mainly applied to Al, Mg, Ti, and other alloys. At present, there are few reports about MAO technology on the RHEAs. We hope that this study can provide a reference for the research on surface modification of RHEAs under high temperature environment. In addition, there have been a lot of reports on the mechanical properties of other types of RHEAs, and this paper focuses on the high temperature oxidation resistance. In this paper, AlTiCrVZr RHEA is selected out of consideration of the characteristics of MAO technology. We prefer to obtain coatings with the composition of Al2O3, TiO2, and Cr2O3, which are favorable for high temperature oxidation resistance.
3. I cannot also accept the statement of the authors: “Micro arc oxidation (MAO) is a kind of surface modification technology with simple equipment, green environmental protection and good economy………”. Comparing to traditional DC anodization MAO requires rather complicated and expensive equipment and does not offer any advantages in comparison with traditional methods from the point of view of environment protection.
Thank you for your question. MAO technology does have some shortcomings as you mentioned. The description in the article may be inaccurate, and we have recently revised it. We believe MAO is environmentally friendly because the electrolyte is weakly alkaline and does not pollute the environment. In addition, all the processes of MAO only need about 10 min, which is suitable for large-scale automatic production. Therefore, we also believe that the technology is high efficiency. We find that many scholars have similar descriptions in their articles, such as “Among them, the micro arc oxidation (MAO) is an effective and ecofriendly surface treatment method, which has attracted much attention”. “As a new surface treatment technology, micro-arc oxidation is simple, efficient and environmentally friendly”. “Micro-arc oxidation is a kind of prevalent surface treatment approach due to its low cost, simple operation as well as environmentally-friendly”.
4. In many places authors present commonly known facts as the conclusions from their investigations. For example, they write: “The elements Si and O are derived from the electrolyte, and Al, Ti, Cr, V, and Zr are from the matrix. This indicating that in the coating formation process of the coating, matrix elements and the solute elements in the electrolyte are involved in the coating-forming reaction.” This is trivial and commonly known. The discussion on the correlation between electronegativity and the susceptibility to oxidation (lines 114 to 124) is trivial, this problem is well known to everyone who studied chemistry, and there is no need to discuss it so widely. It should be shortened to 1 or 2 sentences.
Thank you for your suggestion. we have revised the relevant content in the article and marked it with red font. In line 121 and 169 of the text, add “The less electronegative the element, the greater the affinity for oxygen…”, “It can be seen from the XPS full spectrum that the coating was mainly composed…”.
5. Manuscript requires extensive linguistic correction; many sentences are difficult to be understood. For example, authors write: “The measurement results of the surface roughness of the coating are almost consistent with the results observed in the surface morphology of the coating”. I do not understand what the authors would like to say. What the authors mean: “results observed in the surface morphology of the coating”? What does it mean “almost consistent”?
Thanks for this suggestion of the reviewer, and we have revised the language problems in the article. Meanwhile, we have revised to what you do not understand and marked it with red font in line 138.
6. Particular comments: The numbers on the ordinates in figure 2 are illegible.
Thank you for your question, and we have modified Figure 2.
7. Authors write: “There are micropores of different sizes…” According to UPAC convention micropores are the pores of the diameters of less than 2 nm. Authors should rather write “pores” not micropores”.
Thank you for your suggestion, and we have revised the corresponding parts in the article.
Round 2
Reviewer 1 Report
can be published in the present form
Reviewer 3 Report
Authors answered to all my question. No more objections.